# Prevalence and Clinical Significance of Potential Drug–Drug Interactions in Hospitalized Pediatric Oncology Patients: A Prospective Pharmacoepidemiologic Study

**DOI:** 10.3390/cancers17183054

**Published:** 2025-09-18

**Authors:** Omid Reza Zekavat, Narjes Zarsanj, Adel Sadeghdoust, Alekhya Lavu, Mohammadreza Bordbar, Sherif Eltonsy, Payam Peymani

**Affiliations:** 1Hematology Research Centre, Nemazee Hospital, Shiraz University of Medical Sciences, Shiraz 71937-11351, Iran; ozekavat@sums.ac.ir (O.R.Z.); n.zarsanj@yahoo.com (N.Z.); bordbarm@sums.ac.ir (M.B.); 2Health Policy Research Center, Institute of Health, Shiraz University of Medical Sciences, Shiraz 71348-45794, Iran; m.sadeghdoust@gmail.com; 3College of Pharmacy, Rady Faculty of Health Sciences, University of Manitoba, Winnipeg, MB R3E 0T5, Canada; lavua@myumanitoba.ca (A.L.); sherif.eltonsy@umanitoba.ca (S.E.)

**Keywords:** drug-drug interactions, chemotherapy, pediatric oncology, pharmacoepidemiology, safety

## Abstract

Children with cancer often need many medications at the same time, which can lead to harmful drug interactions. These interactions may make treatments less effective or cause serious side effects. However, there is limited information about such risks in hospitalized children with cancer, especially in the Middle East. In this study, we looked at prescriptions for children receiving cancer treatment in a hospital in Iran. We used two trusted tools to check for possible drug interactions. Our goal was to find out how often these interactions happen, how serious they are, and which drug combinations are most involved. The results can help doctors and pharmacists better manage medications and protect young patients from preventable problems during treatment.

## 1. Introduction

Pediatric cancers represent a significant global health burden, ranking as the second leading cause of death among children aged 5 to 14 years [1]. Advances in multimodal therapy—including chemotherapy—have dramatically improved survival rates in high-income countries, yet treatment remains complex and fraught with risks related to both malignancy and the aggressive therapies employed [2].

Optimal management of pediatric cancer requires the concomitant use of multiple pharmacologic agents to treat the underlying disease and to manage associated symptoms and complications, resulting in high rates of polypharmacy in this vulnerable population. The challenge of polypharmacy in pediatric oncology is compounded by unique developmental considerations: children exhibit age-dependent variability in drug absorption, metabolism, distribution, and excretion [3,4,5]. These pharmacokinetic and pharmacodynamic differences distinguish pediatric patients from adults and may amplify both the risk and the clinical severity of drug–drug interactions (DDIs) [6,7]. Clinical consequences of DDIs in this setting can be severe, including increased toxicity, reduced chemotherapeutic efficacy, unanticipated adverse events, and even life-threatening complications [6,8]. In addition, supportive care medications such as antiemetics, antimicrobials, and analgesics further elevate the risk of clinically significant DDIs [9,10]. Despite the recognized importance of DDIs in pediatric oncology, evidence remains surprisingly limited, particularly from low- and middle-income countries and the Middle East. Existing prevalence estimates range broadly, from 3.8% to 75% in pediatric settings, depending on the patient population, hospital characteristics, and DDI identification methods [3,4]. Most available data are derived from retrospective studies or from adult or mixed-age cohorts, making it difficult for clinicians to generalize findings for pediatric oncology inpatients [11,12,13]. Compounding this knowledge gap is the lack of direct comparison between medication-interaction screening resources, despite widespread clinical reliance on databases such as Lexi-Interact™ and Drugs.com™ for DDI identification [14,15]. Given these challenges, there is an urgent need for prospective, region-specific data characterizing the prevalence, clinical significance, and risk factors for DDIs in hospitalized children with cancer [16].

The present study aims to address this gap by (1) prospectively evaluating the prevalence and clinical significance of potential DDIs (PDDIs) in hospitalized pediatric oncology patients using dual database screening, and (2) characterizing the most common PDDIs and implicated drug pairs in a leading tertiary referral center in Iran. By providing robust, context-specific evidence, our findings are intended to inform clinical practice, support safer polypharmacy, and guide future pharmacovigilance initiatives in regional and global pediatric oncology care.

## 2. Materials and Methods

### 2.1. The Study Design and Setting

This prospective observational study was carried out at the pediatric oncology ward of Amir Oncology Hospital, a major tertiary care referral center affiliated with Shiraz University of Medical Sciences, Shiraz, Iran. The study was conducted over an eight-month period, from November 2019 to June 2020. The hospital treats a broad spectrum of pediatric oncology cases and follows contemporary inpatient chemotherapy protocols as per regional and international standards. Ethical approval was obtained from the university’s institutional review board (approval code IR.SUMS.REC.1399.218; 1 May 2019). All study procedures were conducted in accordance with the Declaration of Helsinki and national regulations governing research involving human participants. Written informed consent was collected from the legal guardians of all patients prior to enrollment.

### 2.2. Study Population

Eligible participants included children and adolescents aged 18 years or younger with a confirmed diagnosis of malignancy who were admitted for the administration of chemotherapy or related oncologic care. Inclusion required that patients were prescribed at least two different medications (including at least one antineoplastic agent) during hospitalization, which allowed for DDI assessment. Exclusion criteria were:Incomplete medical or medication recordsPresence of severe comorbidities that could confound DDI assessment (e.g., advanced congenital anomalies, end-stage organ failure)Refusal or inability to provide informed consent

A consecutive sampling approach was applied to recruit all eligible admissions during the study window. Demographic and clinical data collected at baseline included age, sex, underlying malignancy, disease stage, comorbid conditions, and reason for admission.

### 2.3. Data Collection Procedures

Each patient’s medication regimens were prospectively monitored throughout their hospital stay. Dedicated case report forms (CRFs) were used to systematically record:All prescribed medications, including dosage, frequency, route of administration, and therapeutic classification (e.g., antineoplastic, supportive care, adjunctive therapy)Changes to medication orders during hospitalizationRelevant laboratory and clinical monitoring parametersAdverse drug reactions possibly attributable to DDIs

Medical charts, physician order sheets, and pharmacy dispensing records were reviewed daily until patient discharge.

### 2.4. Drug–Drug Interaction Assessment

Potential drug–drug interactions (PDDIs) were evaluated for every combination of concurrently prescribed medications using two independent, widely used online drug interaction databases: Lexi-Interact™ (https://www.uptodate.com/drug-interactions, access on 1 June 2019) and Drugs.com™ (https://www.drugs.com/drug_interactions.html, access on 1 June 2019). Both tools provide evidence-based information on the mechanisms, severity, and clinical management of DDIs (Appendix A) [17,18]. Concurrent therapy was defined as two medications being prescribed at the same time or with overlapping dates during the hospital stay. We counted DDIs based on the number of patients with at least one DDI, not by treatments or episodes. The main unit is the patient. For each identified DDI:The pair of drugs involved was recorded with generic names.The severity of each interaction was documented according to both Lexi-Interact™ (categories A, B, C, D, X) and Drugs.com™ (classified as minor, moderate, or major).The presumed mechanism (pharmacokinetic vs. pharmacodynamic) was noted.

Only interactions flagged by at least one database were included for analysis; where there was discordance in severity ratings, the higher severity classification was documented to prioritize patient safety.

Interactions were further categorized as antineoplastic–antineoplastic, antineoplastic–non-antineoplastic, or non-antineoplastic–non-antineoplastic.

The clinical team and study pharmacists reviewed all flagged PDDIs to assess their relevance and whether action (e.g., dose adjustment, monitoring) was taken. Causality between ADRs and PDDIs was assessed based on clinical judgment without the use of a standardized tool such as the Naranjo scale.

### 2.5. Outcomes and Definitions

The primary outcome measure was the prevalence of patients with at least one identified PDDI during hospitalization. Secondary outcomes included:Total number of PDDIs per patientDistribution and frequency of DDI drug pairsSeverity grading of observed PDDIsClinical interventions taken in response to PDDIs (when applicable)Incidence of adverse drug reactions linked to identified PDDIs

The prevalence of DDIs was defined as:(1)$$[\frac{Number\;of\;patients\;with\;at\;least\;one\;DDI}{Total\;number\;of\;patients\;enrolled}]\times 100$$

### 2.6. Statistical Analysis

Data were entered and managed using IBM SPSS Statistics version 26.0.2.0 software and independently verified for accuracy. Descriptive statistics were calculated to summarize demographic and treatment characteristics, presented as means (with standard deviations), medians (with ranges), or frequencies (with percentages) as appropriate. The prevalence of PDDIs was calculated as the proportion of patients experiencing at least one PDDI relative to the total study cohort. The most frequent DDI pairs and their severities were tabulated. Differences in PDDI prevalence by demographic or clinical characteristics were explored using appropriate statistical tests (e.g., chi-square or Fisher’s exact test for categorical variables). Sensitivity analyses were performed using an alternative cut-point of ≥5 medications to define polypharmacy, based on prior studies. All analyses were conducted using IBM SPSS Statistics version 26.0.2.0, and a two-sided significance threshold of *p* < 0.05 was used for hypothesis testing.

## 3. Results

### 3.1. Patient Characteristics

A total of 120 hospitalized pediatric oncology patients were screened during the study period; 80 patients met the eligibility criteria and were enrolled. The mean age of patients was 8.9 ± 4.6 years (range: 1–17 years), and 58.8% (*n* = 47) were male. Acute lymphoblastic leukemia (ALL) was the most prevalent malignancy, accounting for 31% (*n* = 25) of cases, followed by acute myeloid leukemia (AML; 10%), non-Hodgkin lymphoma (NHL; 8%), osteosarcoma (8%), and medulloblastoma (6%). A small proportion of patients had rare malignancies including ependymoma, adrenocortical tumor, and ameloblastic carcinoma. Comorbidities were minimal—91% (*n* = 73) of patients had no associated chronic conditions. Among the few patients with comorbidities, diabetes mellitus (2.5%) and genetic disorders such as Down syndrome and ataxia-telangiectasia (each <1.5%) were recorded. Most hospitalizations were scheduled (62.5%) and related to chemotherapy administration (85%), while 11% were due to febrile neutropenia. Detailed patient characteristics are presented in Table 1.

### 3.2. Medication Use Patterns

During hospitalization, patients were prescribed a wide range of therapeutic agents consistent with standard oncology protocols. The median number of medications per patient was 8 (range: 2–14), reflecting the complexity of cancer treatment regimens in pediatric care. The most frequently prescribed drugs were Ranitidine: 66%, Allopurinol: 59%, Granisetron: 53%, Cytarabine (Cytosar): 37%, Methotrexate: 34%, Acetaminophen (Paracetamol): 33% and Co-trimoxazole: 33%. Other commonly used medications included vincristine, etoposide, G-CSF, and ursodeoxycholic acid. Antineoplastic drugs made up nearly half of all prescriptions, while the other half included supportive agents (antiemetics, antibiotics, gastroprotective drugs, etc.), underscoring the polypharmacy burden across this study. Table 2 provides detailed information on the frequency and therapeutic groups of the prescribed drugs administered to the patients.

### 3.3. Prevalence and Types of Potential Drug–Drug Interactions

Potential drug–drug interactions (PDDIs) were identified in 21.2% (*n* = 17/80) of patients, totaling 197 interaction events. The interactions were categorized by drug class combinations (antineoplastic–antineoplastic, antineoplastic–non-antineoplastic, and non-antineoplastic–non-antineoplastic). The distribution of interactions was as follows: Antineoplastic–Antineoplastic interactions: 108 cases (54.8%), non-antineoplastic–non-antineoplastic interactions: 61 cases (30.9%), and Antineoplastic–Non-antineoplastic interactions: 28 cases (14.3%). The frequency of the most common medication pairs involved in potential DDIs is provided in Table 3. Sensitivity analyses using ≥5 medications showed a similar significant association with PDDI risk. Additionally, in about 15% of cases, the two screening tools did not agree on severity classification (Appendix A).

A more detailed assessment of each drug combination and its potential for interaction, as classified by both Lexi-Interact™ and Drugs.com™, is presented in Table 4, providing key insight into the frequency and clinical relevance of DDIs observed in this pediatric oncology cohort. To facilitate interpretation, the table includes drug pairs, frequency, interaction type (pharmacokinetic or pharmacodynamic), severity ratings from each database, and observed adverse drug reactions where applicable. A summary comparison of severity ratings for frequently observed drug pairs across Lexi-Interact™ and Drugs.com™ is provided in Appendix A to highlight areas of agreement and discrepancy between the two databases. The majority of interactions were classified as moderate in severity: 178 interactions (90.4%) were rated as moderate, 17 (8.6%) as minor, and only 2 (1.0%) as major, based on Lexi-Interact™ and Drugs.com™ criteria. The most frequently identified interaction was between acetaminophen and granisetron, found in 61 cases. Although categorized as moderate risk, this combination has been postulated to reduce the analgesic efficacy of acetaminophen due to serotonin antagonism. Among antineoplastic agents, the combination of methotrexate and vincristine was most common, observed in 53 cases. Notably, methotrexate alone was involved in 156 interactions, underscoring its central role in polypharmacy-associated risks in pediatric oncology.

### 3.4. Clinical Outcomes, Adverse Reactions, and Management

Adverse drug reactions (ADRs) potentially attributable to PDDIs were documented in 14 patients (17.5%), with common manifestations including gastrointestinal side effects and hematologic abnormalities. ADRs are considered linked to DDIs based on clinical judgment of temporal association and pharmacological plausibility. Notably, 5% of patients underwent dose modification or enhanced monitoring in response to identified PDDIs during hospitalization. Dose modifications or monitoring were made by treating physicians as part of routine care; the study was purely observational. The distribution of ADRs by type and severity is illustrated in Figure 1, with breakdown by sex and malignancy type. Of the 197 identified interactions, 113 (57.4%) were pharmacodynamic and 84 (42.6%) were pharmacokinetic in nature. However, among the 14 observed ADRs, 10 (71.4%) were linked to pharmacodynamic interactions and 4 (28.6%) to pharmacokinetic interactions, suggesting a greater clinical impact of pharmacodynamic mechanisms in this cohort. All observed ADRs resolved with supportive care or medication adjustments, and no fatal events were directly linked to PDDIs during the study.

### 3.5. Subgroup and Exploratory Analyses

Stratified analysis did not demonstrate statistically significant associations between gender, age group, or malignancy type and the presence of PDDIs (*p* > 0.05). However, a clear trend was observed: patients who were prescribed more than 8 medications during hospitalization were at significantly higher risk of experiencing at least one DDI (*p* < 0.01), emphasizing the importance of polypharmacy monitoring as a clinical risk factor.

Additionally, comparing the Lexi-Interact™ and Drugs.com™ tools revealed slight discrepancies in DDI classification. In approximately 15% of cases, severity level ratings differed, with Lexi-Interact™ tending to classify interactions as more clinically relevant in borderline cases.

## 4. Discussion

This prospective study provides comprehensive data on potential drug–drug interactions (PDDIs) in hospitalized pediatric oncology patients in Iran, representing the largest such investigation in the region. Our findings reveal that 21.2% of children under active oncology care experienced at least one PDDI during their hospital admission. This rate, while substantial, is somewhat lower than the upper bounds reported in previous studies of pediatric and mixed-age cancer cohorts, where PDDI prevalence has ranged widely—from as low as 3.8% to over 70% depending on methodology, population, and detection tools [19,20,21]. Notably, patients with at least one PDDI experienced a high cumulative burden, averaging over 11 interactions per patient, reflecting the complexity of polypharmacy in pediatric oncology care. Several factors may account for such variation, including differences in oncology protocols, patient characteristics, and, crucially, the drug interaction detection resources used. The dual-database approach (Lexi-Interact™ and Drugs.com™) applied in this study marks a methodological strength and directly addresses limitations in the existing literature, which has most often relied on a single-source DDI checker. The dual screening captured a broader range of interaction alerts, and notable yet clinically relevant discrepancies in severity grading between the two databases signal the importance of multi-database verification, especially in high-risk settings such as pediatric oncology. However, we did not apply a formal causality tool (e.g., Naranjo scale) to assess the link between ADRs and PDDIs, which may affect the consistency of attribution.

Methotrexate and vincristine were the most frequently implicated antineoplastic agents in PDDIs. Both are core components of standard pediatric leukemia protocols, and their combined use increases the risk for additive toxicity, particularly hepatotoxicity and neurotoxicity [22,23,24,25]. The most common non-antineoplastic DDI was observed between acetaminophen and granisetron. Recent pharmacologic evidence suggests serotonin antagonists (like granisetron) may attenuate the analgesic effects of acetaminophen, an interaction not always top-of-mind for pediatric oncology providers but relevant for supportive care [26,27].

Importantly, the majority of detected PDDIs were rated as moderate in clinical significance. Nevertheless, even moderate interactions, if left unchecked, can result in suboptimal treatment outcomes key concerns in immunocompromised and medically complex children [28]. The relatively low percentage of major-risk PDDIs observed aligns with previous reports but should not deter implementation of robust clinical management strategies [19,20]. In this study, 5% of patients required modification of regimen or enhanced monitoring as a direct result of identified PDDIs, underscoring the real-world impact of systematic medication review. Adverse drug reactions attributable to DDIs were infrequent and generally resolved with supportive management and/or drug adjustment. A 2015 study reported that 5% of hospitalized pediatric patients experienced contraindicated PDDIs, aligning with our finding that 5% required regimen changes or enhanced monitoring [19]. Similarly, a 2021 PICU study found clinically relevant PDDI-related events in 10.1% of patients, highlighting the value of systematic medication review [29]. Notably, higher levels of polypharmacy (more than eight concurrent medications) were associated with a greater risk of experiencing at least one DDI, emphasizing the practical need for heightened vigilance in patients with more complex medication regimens [8,30,31]. The association persisted when applying the alternative literature-based threshold, supporting robustness. Strengths of this work include its prospective design, systematic and blinded data collection, application of dual DDI databases, and its focus on a previously understudied population in the Middle East. Limitations include the single-center setting, which may limit generalizability, and a sample size that, while large for the region, may not reflect the full diversity of pediatric malignancies and supportive care requirements elsewhere. In addition, pharmacokinetic variables such as renal and hepatic function or pharmacogenomic data were not systematically analyzed but may influence the clinical expression of DDIs. In addition, pharmacokinetic variables such as renal and hepatic function or pharmacogenomic data were not systematically analyzed but may influence the clinical expression of DDIs. As this was a descriptive study, multivariable analysis was not performed, which limits interpretation of potential confounders and should be addressed in future research.

Future research should prioritize multicenter, multinational collaborations incorporating clinical outcome adjudication, pharmacovigilance initiatives, and interventional studies aimed at minimizing DDI-related harm. Integration of electronic health records with automated, multi-database DDI alerts and routine clinical pharmacist involvement may further reduce the risk to this vulnerable population [32].

## 5. Conclusions

Potential drug–drug interactions are a common and clinically relevant concern among hospitalized pediatric oncology patients, particularly in regions with complex, multi-agent chemotherapy protocols. The use of dual DDI screening databases enhances detection and risk stratification. Methotrexate, vincristine, and combinations of supportive care medications require particular scrutiny. Routine, multidisciplinary review of prescribed regimens—supported by robust electronic or clinical pharmacist-led systems—is strongly recommended to optimize safety and treatment efficacy. These findings highlight the need for context-specific DDI management strategies and serve as a foundation for improving drug safety in pediatric oncology practice in Iran and comparable settings.

## Figures and Tables

**Figure 1 cancers-17-03054-f001:**
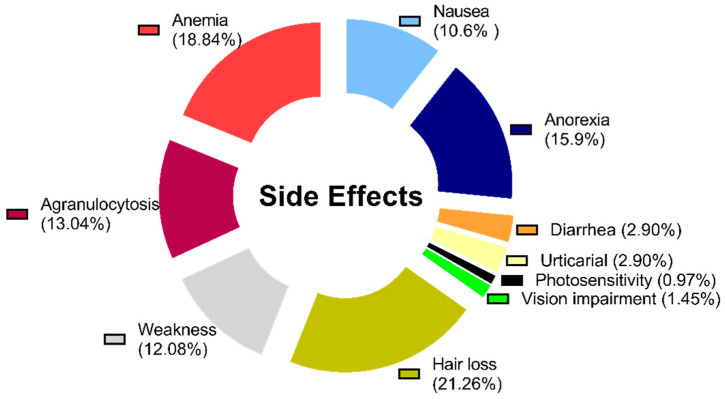
The donut graph shows the percentage of observed adverse drug reactions. The distribution by gender is 39% female and 61% male.

**Table 1 cancers-17-03054-t001:** Patients’ demographic and clinical characteristics.

Variable			Patients (*n* = 80) *
Age, y			8.9 ± 4.6 range (1–17)
Gender	Female		33 (41.3%)
	Male		47 (58.8%)
Weight (kg)			30.2450
Height (cm)			128.9625
Malignancy type			
ALL			25 (31.0%)
AML			8 (10.0%)
HL			2 (2.0%)
NHL			7 (8.0%)
Neuroblastoma			1 (1%)
Osteosarcoma			7 (8%)
Medulloblastoma			5 (6%)
Ewing sarcoma			6 (7%)
Hepatoblastoma			4 (5%)
Wilm’s tumour			4 (5%)
Soft tissue sarcoma			2 (2%)
Rhabdomyosarcoma			3 (3%)
GCT			2 (2%)
Mixed line leukemia			1 (1%)
Ependymoma			1 (1%)
Ameloblastic carcinoma			1 (1%)
Adrenocortical tumor			1(1%)
Comorbidities			
None			73 (91%)
DM			2 (2%)
Ataxia telangiectasia			1(1%)
CP			1(1%)
Down			1(1%)
Minor thalassemia			1 (1%)
Myelodysplastic syndrome			1(1%)
Admission cause			
Chemotherapy			68 (85%)
Febrile neutropenia			9 (11%)
Chemotherapy & Neutropenia			2 (2%)
Meningitis			1(1%)
Chemotherapy		Yes	72(90%)
		No	8 (10%)
Admission		Urgent	30 (37.5%)
		Scheduled	50 (62.5%)

* Data are presented as mean ± standard deviation or number (percentage). Abbreviations: ALL, acute lymphoblastic leukemia; AML, acute myeloid leukemia; HL, Hodgkin lymphoma; NHL, non-Hodgkin lymphoma.

**Table 2 cancers-17-03054-t002:** The frequency and therapeutic group of drugs prescribed to patients.

Drug Name	*n* (%)	Therapeutic Group ^1^	Drug Name	*n* (%)	Therapeutic Group ^1^
Ranitidine ^2^	66	A02BA	Cyclophosphamide	8	L01AA
Allopurinol	59	M04AA	Ifosfamide	11	L01AA
Granisetron	64	A04AA	Amphotericin B	7	J02AA
Cytosar	37	L01BC	Vancomycin	7	J01XA
Methotrexate	34	L04AX	Irinotecan	5	L01CE
Paracetamol	33	N02BE	Pantoprazole	5	A02BC
Co-Trimoxazole	33	J01EE	Mercaptopurine (6-MP)	5	L01BB
Vincristine	29	L01CA	Voriconazole	5	J02AC
Ursodeoxycholic acid	25	A05AA	Bevacizumab	4	L01FG
Etoposide	22	L01CB	Clindamycin	4	J01FF
G-CSF	22	L03AA	Doxorubicin	4	L01DB
Dexamethasone	21	H02AB	Piperacillin/tazobactam	4	J01CR
Mesna	19	R05CB	Amikacin	3	J01GB
Meropenem	18	J01DH	Bleomycin	3	L01DC
Ciprofloxacin	17	J01MA	Cefazolin	3	J01DB
Hydrocortisone	16	H02AB	Ondansetron	3	A04AA
Livergol^®^ ^3^	14	A05BA	Pantoprazole	3	A02BC
Doxorubicin	13	L01DB	Metoclopramide	3	A03FA
Pegaspargase	13	L01XX	Rituximab	3	L01FA
Cisplatin	12	L01XA	Temozolomide	3	L01AX
Leucovorin	12	V03AF	Fluorouracil (5-FU)	2	L01BC

^1^ Therapeutic Group by Anatomical Therapeutic Chemical (ATC) Classification system. ^2^ Ranitidine was included based on its availability in the local formulary during the study period. We acknowledge that this agent has since been withdrawn in many countries due to NDMA contamination concerns. ^3^ Preparations containing silibinin, a standardized extract of milk thistle seeds.

**Table 3 cancers-17-03054-t003:** The frequency of most common pairs of drugs with potential drug–drug interaction.

Pairs of Drugs	*n* (%)	Pairs of Drugs	*n* (%)	Pairs of Drugs	*n* (%)
Acetaminophen–Granisetron	26 (13.1%)	Cisplatin–Etoposide	4 (2%)	Etoposide–Cytarabine	1 (0.5%)
MTX–VCR	16 (8.1%)	MTX–Idarubicin	4 (2%)	Adrianycin–Etoposide	1 (0.5%)
MTX–Cotrimoxazole	16 (8.1%)	Doxorubicin–Cisplatin	3 (1.5%)	Cotrimoxazole–Fluconazole	1 (0.5%)
Pegaspar–VCR	14 (7.1%)	Bleomycin–Etoposide	3 (1.5%)	Cotrimoxazole–Linezolid	1 (0.5%)
Cotrimoxazole–Ciprofloxacin	13 (6.5%)	VCR–Cisplatin	3 (1.5%)	Pegaspar–Cytarabine	1 (0.5%)
Pegaspar–MTX	11 (5.5%)	Bleomycin–Cisplatin	2 (1%)	Fluconazole–Granisetrone	1 (0.5%)
MTX–Ciprofloxacin	10 (5%)	CCNU–Cisplatin	2 (1%)	Fluconazole–Ciprofloxacin	1 (0.5%)
MTX–Leukoverin	10 (5%)	VCR–Fluconazole	1 (0.5%)	Oxaliplatin–Etoposide	1 (0.5%)
MTX–Adriamycin	10 (5%)	Ranitidine–Fluconazole	1 (0.5%)	Cisplatin–Cytarabine	1 (0.5%)
Etoposide–Cytozar	8 (4%)	Irinotecan–Sirolimus	1 (0.5%)	Azithromycin–Fluconazole	1 (0.5%)
Ciprofloxacin–Hydrocortisone	7 (3.5%)	Linezolid–Granisetrone	1 (0.5%)	Dexamethasone–Fluconazole	1 (0.5%)
Ciprofloxacin–Dexamethasone	6 (3%)	MTX–Cytarabine	1 (0.5%)	Ranitidine–Azithromycin	1 (0.5%)
MTX–6MP	4 (2%)	Cytarabine–Adriamycin	1 (0.5%)	Cytarabine–Idarubicin	1 (0.5%)
5FU–Leukoverin	4 (2%)	MTX–Etoposide	1 (0.5%)	MTX–Azithromycin	1 (0.5%)

**Table 4 cancers-17-03054-t004:** The drug combinations with the potential to interact according to LEXI interact and Drugs.com.

Chemotherapy Regimen		LEXI Interact	Drugs.com	Other Drugs (LEXI)	LEXI Interact
1	Etoposide Cytosar	No	Moderate ^1^	No	N.A.
2	Methotrexate Leucovorin	A	Moderate; Leucovorin may reduce the effects of methotrexate.	Methotrexate & Azithromycin	C: Azithromycin may increase the serum concentration of Methotrexate
3	Vincristine Irinotecan Temozolomide Bevacizumab	No	No	Vincristine & Fluconazole	C: Neuropathies, gastrointestinal toxicities
				Acetaminophen & Granisetron	B: Granisetron may diminish the analgesic effect of acetaminophen
				Fluconazole & ranitidine	A
4	Vincristine Mesna Ifosfamide Etoposide	No	No	Ciprofloxacin & Dexamethasone	C: Tendonitis, tendon rupture
				Ciprofloxacin & Cotrimoxazole	C: Hypoglycemia
5	Irinotecan Prednisolone Sirolimus	No	Irinotecan & Sirolimus (moderate); Using sirolimus together with irinotecan may increase the blood levels and effects of one or both medications.	Linezolide & Granisetron	C: Hyperreflexia, clonus, hyperthermia, diaphoresis, tremor, autonomic instability, mental status change
6	Etoposide Adriamycin Cytarabine Methotrexate	No	1- Methotrexate & Cytarabine (Moderate); Liver and/or nervous system problems. 2- Methotrexate & Adriamycin (Moderate) ^1^ 3- Adriamycin & Cytarabine (Moderate) ^1^ 4- Methotrexate & Etoposide (Moderate) ^2^ 5- Etoposide & Cytarabine (Moderate) ^2^ 6- Adriamycin & Etoposide (Minor).	No	N.A.
7	Vincristine PegAspar Methotrexate Cytarabine	No	1- Methotrexate & Cytarabine (moderate); Liver and/or nervous system problems. 2- Methotrexate & Vincristine (moderate); Liver problems. 3- PegAspar & Methotrexate (moderate); Pegaspargase may reduce the effects of methotrexate in the treatment of some conditions. 4- PegAspar & Cytarabine (moderate); Liver damage. 5- Vincristine & PegAspar (moderate); Liver damage	Methotrexate & Cotrimoxazole	D: Bone marrow suppression
				Ciprofloxacin & Hydrocortisone	
				Linezolide & Granisetron	C: Hyperreflexia clonus, hyperthermia, diaphoresis, tremor, autonomic instability, mental status change.
				Methotrexate & Ciprofloxacin	C: Ciprofloxacin may increase the serum concentration of methotrexate.
				Cotrimoxazole & Ciprofloxacin	C: Hypoglycemia
				Cotrimoxazole & Fluconazole	B: Fluconazole may increase the serum concentration of Sulfamethoxazole
				Cotrimoxazole & Linezolide	C: Hypoglycemia
				Vincristine and Fluconazole	C: Neuropathies, gastrointestinal toxicities
				Acetaminophen & Granisetron	B. Granisetron may diminish the analgesic effect of acetaminophen.
				Fluconazole & ciprofloxacin	B: QTc-prolonging effect
				Fluconazole & Granisetron	B: QTc-prolonging effect
				Fluconazole & ranitidine	A
8	Oxaliplatin Etoposide Bevacizumab	No	Oxaliplatin & Etoposide (moderate); Nerve damage.	Dexamethasone & Granisetron	D
9	Doxorubicin Cisplatin	No	Moderate ^1^	Doxorubicin conventional & Granisetron	D
				Dexamethasone & Granisetron	D
				Doxorubicin liposomal & Granisetron	C
10	Cisplatin Vincristine 5-fluorouracil Leucovorin	Fluorouracil and leucovorin (C): Diarrhea, Mucositis/stomatitis, neutropenia	Major; Anemia, bleeding problems, infections, and nerve damage (fluorouracil and leucovorin).	No	N.A.
11	Methotrexate 6-Mercaptopurine	No	Minor	Granisetron & Hydrocortisone	D
12	Cisplatin Etoposide Bleomycin	No	1- Bleomycin & cisplatin (moderate); Nerve damage. 2- Bleomycin & Etoposide (moderate) ^1^; 3- Cisplatin and Etoposide (moderate); Nausea and vomiting, fever, chills, sore throat, flu symptoms, easy bruising, or extreme weakness.	No	N.A.
13	Cisplatin Etoposide Cytarabine	No	1- Cisplatin & Cytarabine (moderate) ^1^ 2- Cisplatin & etoposide (moderate); Nausea and vomiting, fever, chills, sore throat, flu symptoms, easy bruising, or extreme weakness 3- Cytarabine & Etoposide (moderate) ^1^	Azithromycin & Fluconazole	C: Monitor for QTc interval prolongation and ventricular arrhythmias
				Ciprofloxacin & Hydrocortisone	C: Tendonitis, tendon rupture
				Dexamethasone & Fluconazole	C: Fluconazole may decrease the metabolism of Dexamethasone
				Linezolide & Granisetron	C: Hyperreflexia, clonus, hyperthermia, diaphoresis, tremor, autonomic instability, mental status change
				Ranitidine & Azithromycin	C
				Cotrimoxazole & Ciprofloxacin	C: Hypoglycemia
				Cotrimoxazole & fluconazole	C
				Cotrimoxazole & Linezolide	C: Hypoglycemia
				Acetaminophen & Granisetron	B: Granisetron may diminish the analgesic effect of acetaminophen.
				Fluconazole & Ciprofloxacin	B: QTc-prolonging effect
				Fluconazole & Granisetron	B: QTc-prolonging effect
				Fluconazole & Ranitidine	A
14	Cisplatin Vincristine 5-fluorouracil Leucovorin	Fluorouracil and Leucovorin (C): *Diarrhea, mucositis/Stomatitis, neutropenia*	Fluorouracil & leucovorin (Major); Anemia, bleeding problems, infections, and nerve damage.	No	N.A.
15	6-Mercaptopurine Methotrexate Leucovorin	Methotrexate and Leucovorin (A)	1- Methotrexate & Leucovorin (Moderate); Leucovorin may reduce the effects of methotrexate. 2- Methotrexate and Mercaptopurine (Minor);	No	N.A.
16	Lomustine Vincristine Cisplatin	No	1- Vincristine & Cisplatin (Moderate); Nerve damage. 2- Cisplatin & Lomustine (Moderate) ^1^	Vincristine & Granisetron	C
					A
17	Idarubicin Cytarabine Methotrexate	No	1- Methotrexate & Cytarabine (Moderate); Liver and/or nervous system problems. 2- Methotrexate & Idarubicin (Moderate) ^1^ 3- Cytarabine & Idarubicin (Moderate) ^1^	No	N.A.

^1^ Nausea, vomiting, diarrhea, loss of appetite, mouth sores, abdominal pain, delayed wound healing, and impaired bone marrow function. ^2^ Loss of appetite; mouth sores; abdominal pain; delayed wound healing; and impaired bone marrow function.

## Data Availability

The original contributions presented in the study are included in the article/Appendix A; further inquiries can be directed to the corresponding author.

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
