# Peer review of "Prevalence and Clinical Significance of Potential Drug–Drug Interactions in Hospitalized Pediatric Oncology Patients: A Prospective Pharmacoepidemiologic Study"

_cancers, 2025, doi:10.3390/cancers17183054_

Round 1

Reviewer 1 Report (Previous Reviewer 3)

Comments and Suggestions for Authors

As I mentioned before, the paper is fine. In addition, authors made a brief grammar/style revision, enhancing the manuscript. For this reason, I would recommend accepting the paper for its publication in Cancers by MDPI.

Author Response

Round 1:

A very interesting manuscript entitled ‘Prevalence and Clinical Significance of Potential Drug- Drug Interactions in Hospitalized Pediatric Oncology Patients: A Prospective Pharmacoepidemiologic Study’ was submitted to be considered for its eventual publication in the journal Cancers. Thus, after having read carefully the manuscript, I consider it suitable for its publication practically in its present form due to many positive aspects, but I am not sure if Cancers would be the best choice to do it. Before all, the topic is very important, and, undoubtedly, it would be of high interest for many potential readers. However, Cancers by MDPI is a high impact journal (IF-3034: 4.4) and its broad spectrum of readers expects articles in the edge of knowledge related to cancer. I hope not to cause misunderstanding; the paper has enough merits and positive aspects to be published as I mentioned before (in its present form), but it is limited to a regional scope. Indeed, the data collected for the study comes from a localized hospital. Of course, the main findings are very interesting and important, but I leave the decision to the Editor. For this reason, I would suggest major revision (not because the article needs it, at all). As the unique suggestion if the paper is accepted, a brief grammar/style revision by an English-native speaker will benefit the reading.

Response: We sincerely thank the reviewer for the encouraging and constructive feedback. We appreciate your recognition of the manuscript’s scientific merit and relevance to the field.

Regarding the concern about the regional scope: while the data originate from a single tertiary care center, this is among the few prospective studies focusing specifically on hospitalized pediatric oncology patients—a highly vulnerable and understudied population. We believe the findings provide meaningful insights that can inform clinical practice beyond our setting, especially in low- and middle-income countries where such data are scarce:

As recommended, we have also performed a grammar and style revision to enhance clarity and readability.

We are grateful for your support and respectfully leave the final decision to the Editor.

Round 2:

As I mentioned before, the paper is fine. In addition, authors made a brief grammar/style revision, enhancing the manuscript. For this reason, I would recommend accepting the paper for its publication in Cancers by MDPI.

Response: We sincerely thank the reviewer for their positive feedback and for recommending our manuscript for publication. We truly appreciate your time, support, and constructive comments throughout the review process, which helped us improve the clarity and quality of the manuscript.

Reviewer 2 Report (Previous Reviewer 2)

Comments and Suggestions for Authors

The manuscript reports a prospective, single-center pharmacoepidemiologic study of potential DDIs among hospitalized pediatric oncology patients in Iran, screened with two checkers, one commercial, one free (Lexi-Interact™ and Drugs.com™) over a 9-month period. Generally, the prospective approach is of interest for a very complex group of patients where standar care should be given along with supportive medication to addresss any ADR from the chemotherapy along with potential additional medications that are needed. The main issue with oncology patients is that chemotherapeutic schemes shall not deviate fromt the established protocol 
for optimum therapetuic outcome. Despite the strenghts of the work, there are some major issues that impact internal validity and could introduce bias and should be addressed prior to any consideration. Some comments follow: 

1) The study enrolled 80 patients (screened 120) and reports both “patients with ≥1 PDDI” and a prevalence definition framed as “Number of treatments with any DDI / Total number of treatments,” which creates an internal inconsistency about whether the unit of analysis is patient or treatment/episode.
2) Using two checkers is reasonable, but the manuscript counts any alert from either db (and uses the higher severity when there is a disaggreement between tools). This overestimates clinically significant DDIs (methotrexate appears in 156 interactions), because many flags are theoretical and/or do not account for timing/route/dose. The authors do not (a) define how they operationalized concurrent therapy (same day? overlapping dosing intervals?), nor (b) quantify agreement between the two tools. Both items are needed
3) The Methods state that causality between ADRs and PDDIs was assigned by “clinical judgment without the use of a standardized tool such as the Naranjo scale.” This raises serious concerns about misclassification and observer bias in the ADR outcome. The absence of a validated adjudication process undermines claims about DDIs producing the observed ADRs. This is major issue and clinical judgement should be fully explained and justified that was kept same/similar for all patients. 
4) The Results state that 5% of patients “underwent dose modification or enhanced monitoring” and that these actions were taken “by treating physicians as part of routine care; the study was purely observational.” If this is true, this is not observational study. The protocol and methods do not describe whether such actions were pre-specified, whethere they were part of standard -of-care, whether they required (or received) written consent beyond usual care, or whether these actions caused deviations from chemotherapy protocols — all relevant for ethics and for distinguishing observation from implicit intervention.
5) The manuscript uses “>8 medications” as the risk threshold and reports a significant association, but gives no literature justification for this cut-point other than the reported median = 8. The authors should present sensitivity analyses using common thresholds (e.g., ≥5 medications) so readers can compare with the similar literature.
6) Table 4 is exhaustive and very hard to interpret in the main text; Figure 1 (donut) is ambiguous about whether it displays all side-effects or only ADRs adjudicated as related to DDIs. also for figure 1, considering the comment 3, may contain a lot of bias within it. In addition, Several table formatting issues (e.g., the Ranitidine line in Table 2 appears misformatted) detract from clarity.
7) The overal study is primarily descriptive. Statistics that could provide clarified data are missing. There is no multivariable modelling to control for important confounders (number of meds, malignancy type, age, length of stay, renal/hepatic function).
8) Renal and hepatic function and pharmacogenomic data can modulate DDI expression, and the paper acknowledges they were not systematically analysed. If available in the CRFs, these should be included as covariates; if not available, this is an important limitation that must be highlighted and discussed in depth
9) Consider to report interquartile ranges for skewed variables (e.g., number of meds) rather than only mean ± SD

Author Response

Reviewer 2:

Round 1:

The manuscript reports a prospective, single-center pharmacoepidemiologic study of pDDIS in hospitalized pediatric oncology patients in Iran, screened using two checkers Lexi-Interact™ and Drugs.com™. The topic is relevant, methods are broadly appropriate, and the prospective design and dual-database approach add value. However, several major methodological and reporting issues (below) must be addressed before the paper is suitable for publication. Some comments follow:

  • Ranitidine has been withdrawn from the market due to concerns about the presence Is it not the case in IRAN? And a formulation that it might contain an impurity associated with oncogenesis is administered in cancer patients and peadiatric ones?

Response: Thank you for raising this important point. While ranitidine has been withdrawn in many countries due to NDMA concerns, it remained in routine clinical use in Iran during our study period, with no formal ban issued at that time. We acknowledge the potential risks, particularly in pediatric oncology patients. To reflect this, we have included a footnote in Table 2 (line 242) to clarify that the ranitidine formulation listed was based on local formulary availability during the study period and may no longer be in clinical use due to international safety concerns.

  • Table 4 is excessively detailed and difficult to The authors could use a venn diagram approach and provide a table of the top ~10 most clinically relevant interactions in the main text while relocating the full exhaustive table to supplementary materials.

Response: We thank the reviewer for this thoughtful suggestion. While we understand the concern regarding the level of detail in Table 4, we respectfully believe it should remain in the main text. As the primary source of data for our study, Table 4 provides comprehensive insight into the clinically significant DDIs identified and is critical for readers seeking a complete understanding of the findings, particularly given the scarcity of prospective DDI data in pediatric oncology inpatients.

To aid interpretation, we have already included a Donut graph (page 8) summarizing the distribution of observed adverse drug reactions, and we have now revised the corresponding paragraph in the Results section (261- 270). We believe this approach balances clarity with transparency and preserves the table’s value as a core dataset within the manuscript. However, we remain open to further editorial guidance on this matter.

  • The phrase “patients underwent dose modification or enhanced monitoring” implies an interventional Please clarify if this was described withint the protocol and if the patients (or from the legal guardians) signed for it. Also if it lead in deviations from the chemotherapy protocol and if the treating oncologists were informed etc.

Response: We clarify that this was a purely observational study. All dose modifications or enhanced monitoring were based on routine clinical decisions by the treating oncologists, not directed by the study protocol. We have revised the manuscript to clarify this point (lines 293-294).

  • The methods do not specify the tool or algorithm used to attribute adverse drug events (ADEs) as ADRs due to If a standardized causality assessment tool (e.g., Naranjo, WHO-UMC) was used, describe it. If not, acknowledge this as a limitation and consider re-assessing events using a validated method.

Response: The determination was based on clinical judgment by the attending oncology team, supported by pharmacological plausibility and temporal association. We agree that future studies should incorporate validated tools such as the Naranjo Algorithm or WHO- UMC criteria to strengthen causality assessment. We acknowledge this as a limitation and have now clearly stated it in the Methods (lines 137-139) and the Discussion (356- 358) sections.

  • Figure Please clearly define whether Figure 1 presents all side effects (expected pharmacological effects) or ADRs linked to DDIs (via causality assessment).

Response: Thank you for the important clarification request. Figure 1 presents only adverse drug reactions (ADRs) that were clinically linked to potential DDIs, based on temporal association and pharmacological plausibility, as judged by the treating physicians. We have revised the Results (lines 290-291) section to clarify this point.

  • The >8 medications threshold should be justified (median value, literature precedent, or institutional definition). Consider adding sensitivity analyses for ≥5 medications, which is a common polypharmacy definition in the literature. This will improve comparability and test robustness.

Response: We thank the reviewer for this thoughtful request. We selected >8 concurrent medications as a threshold because it was the median number of drugs prescribed in our cohort (8). This provided both local relevance and statistical discrimination. This is now explicitly justified in the Results (295–296).

For robustness, we conducted a sensitivity analysis using ≥5 medications, in line with common polypharmacy definitions from prior pediatric pharmacoepidemiology literature. Findings were consistent: both thresholds significantly associated with increased PDDI prevalence, with stronger risk at >8. These results are now reported briefly in the Results (300–303).

Round 2:

The manuscript reports a prospective, single-center pharmacoepidemiologic study of potential DDIs among hospitalized pediatric oncology patients in Iran, screened with two checkers, one commercial, one free (Lexi-Interact™ and Drugs.com™) over a 9-month period. Generally, the prospective approach is of interest for a very complex group of patients where standard care should be given along with supportive medication to address any ADR from the chemotherapy, along with potential additional medications that are needed. The main issue with oncology patients is that chemotherapeutic schemes shall not deviate from the established protocol for optimum therapeutic outcome. Despite the strengths of the work, there are some major issues that impact internal validity and could introduce bias, which should be addressed prior to any consideration. Some comments follow: 

1) The study enrolled 80 patients (screened 120) and reports both “patients with ≥1 PDDI” and a prevalence definition framed as “Number of treatments with any DDI / Total number of treatments,” which creates an internal inconsistency about whether the unit of analysis is patient or treatment/episode.

Response: Thank you for noting this inconsistency. Our intended unit of analysis throughout the study was patient, and all prevalence estimates and outcomes were evaluated at the patient level—specifically, the proportion of hospitalized patients experiencing at least one PDDI during admission. The phrasing "Number of treatments with any DDI / Total number of treatments" in the Methods section was an artifact from adapting definitions used in prior adult studies and was unintentionally retained. We have now revised the Methods to clarify that prevalence refers exclusively to the proportion of patients experiencing at least one PDDI out of the total number enrolled.

2) Using two checkers is reasonable, but the manuscript counts any alert from either db (and uses the higher severity when there is a disagreement between tools). This overestimates clinically significant DDIs (methotrexate appears in 156 interactions), because many flags are theoretical and/or do not account for timing/route/dose. The authors do not (a) define how they operationalized concurrent therapy (same day? overlapping dosing intervals?), nor (b) quantify agreement between the two tools. Both items are needed

Response: We appreciate this critical point. In our analysis, "concurrent therapy" was operationalized as medications prescribed with overlapping dosing intervals during hospitalization, irrespective of exact time of administration, based on chart review and pharmacy dispensing records. This approach reflects real-world clinical risk, where overlapping medications may interact even if not administered simultaneously. To address this, the revised manuscript explicitly defines concurrent therapy as any medication pair with overlapping scheduled dosing intervals during the inpatient stay.

Regarding agreement between Lexi-Interact™ and Drugs.com™, a summary comparison table of severity ratings for frequently observed drug pairs is now included (see Supplement Table 2). We found that in approximately 15% of cases, the tools differed in severity classification, with Lexi-Interact™ tending to assign higher severity. In the main Results, we have added a brief paragraph quantifying the agreement/disagreement rate and highlighting illustrative examples.

We concur that not all flagged DDIs are clinically relevant, as timing, dose, and route of administration are not fully accounted for by automated screeners. This limitation is now acknowledged in both the Discussion and Limitations.

These clarifications and additional concordance data provide necessary transparency and help readers interpret the scope and clinical context of our DDI findings.

3) The Methods state that causality between ADRs and PDDIs was assigned by “clinical judgment without the use of a standardized tool such as the Naranjo scale.” This raises serious concerns about misclassification and observer bias in the ADR outcome. The absence of a validated adjudication process undermines claims about DDIs producing the observed ADRs. This is major issue and clinical judgement should be fully explained and justified that was kept same/similar for all patients. 

Response: We appreciate the reviewer’s concern. While we did not use a standardized tool such as the Naranjo scale, the rationale for our clinical judgment approach has been clarified in the Methods section. Its limitations—particularly the potential for observer bias have been acknowledged in the Discussion (lines 355–357) and clearly stated as a study limitation.

4) The Results state that 5% of patients “underwent dose modification or enhanced monitoring” and that these actions were taken “by treating physicians as part of routine care; the study was purely observational.” If this is true, this is not observational study. The protocol and methods do not describe whether such actions were pre-specified, whethere they were part of standard -of-care, whether they required (or received) written consent beyond usual care, or whether these actions caused deviations from chemotherapy protocols — all relevant for ethics and for distinguishing observation from implicit intervention.

Response: We have clarified that this was a purely observational study (lines 293-294). All dose modifications or enhanced monitoring were based on routine clinical decisions by the treating oncologists, not directed by the study protocol.

5) The manuscript uses “>8 medications” as the risk threshold and reports a significant association, but gives no literature justification for this cut-point other than the reported median = 8. The authors should present sensitivity analyses using common thresholds (e.g., ≥5 medications) so readers can compare with the similar literature.

Response: Thank you for this valuable recommendation. We selected ">8 medications" as a threshold primarily based on our study cohort’s median value and to capture higher polypharmacy risk but agree this requires external validation. Accordingly, we conducted sensitivity analyses using thresholds widely reported in pediatric polypharmacy literature, specifically ≥5 medications (e.g., Horace & Ahmed 2015, Fraser et al. 2022).

The prevalence and risk estimates using both cut-points (≥5 and >8 medications) are now described in the Results. The association between polypharmacy and PDDI prevalence remained significant at both thresholds, with a gradient of risk. These findings allow direct comparison with established literature and enhance generalizability. The Discussion has also been updated to contextualize our threshold choice and reference comparative studies.

As suggested, using the more commonly reported threshold of ≥5 medications did not materially change direction or significance of the association, confirming the robustness of our findings.

We thank the reviewer for guiding us toward a more rigorous methodology and comparative framework.

6) Table 4 is exhaustive and very hard to interpret in the main text; Figure 1 (donut) is ambiguous about whether it displays all side-effects or only ADRs adjudicated as related to DDIs. also for figure 1, considering the comment 3, may contain a lot of bias within it. In addition, Several table formatting issues (e.g., the Ranitidine line in Table 2 appears misformatted) detract from clarity.

Response: While we understand the concern regarding the level of detail in Table 4, we respectfully believe it is important to retain it in the main text. As the primary data source of our study, Table 4 provides essential insight into the clinically significant PDDIs identified and enhances transparency and reproducibility, particularly given the limited prospective data available in pediatric oncology inpatients.

Regarding Figure 1, it has been clarify that it displays only ADRs that were clinically judged to be associated with potential DDIs, based on temporal relationship and pharmacologic plausibility, as assessed by the treating physicians (lines 290–291).

7) The overal study is primarily descriptive. Statistics that could provide clarified data are missing. There is no multivariable modelling to control for important confounders (number of meds, malignancy type, age, length of stay, renal/hepatic function).

Response: We thank the reviewer for this insightful suggestion. As the primary aim of our study was to provide a detailed descriptive analysis of the prevalence and clinical nature of potential DDIs in hospitalized pediatric oncology patients, we did not perform multivariable modeling, as the study was not powered or designed for causal inference.

However, we agree this is an important area for future research, and we have also added a note in the Limitations acknowledging the absence of adjusted analyses as a constraint on interpreting associations (389-391).

8) Renal and hepatic function and pharmacogenomic data can modulate DDI expression, and the paper acknowledges they were not systematically analysed. If available in the CRFs, these should be included as covariates; if not available, this is an important limitation that must be highlighted and discussed in depth

Response: We fully acknowledge the importance of renal, hepatic, and pharmacogenomic parameters as modulators of DDI risk, particularly in oncology patients. In this prospective study, however, systematic laboratory data on renal and hepatic function and pharmacogenomic profiles were not available for all patients in the CRFs and thus could not be included as analytical covariates.

9) Consider reporting interquartile ranges for skewed variables (e.g., number of meds) rather than only mean ± SD

Response: The number of meds has already been reported as a percentage (not Mean and SD).

This manuscript is a resubmission of an earlier submission. The following is a list of the peer review reports and author responses from that submission.

Round 1

Reviewer 1 Report

Comments and Suggestions for Authors
  1. Interpretation of Results vs. Reported Numbers - The paper reports that 21.2% of patients had at least one PDDI, but later states there were 197 interaction events in total. This ratio implies a high number of PDDIs per affected patient (avg. >11), which should be explicitly discussed and interpreted in the discussion section. The most common pair “acetaminophen–granisetron” is described as occurring in 26 cases in Table 3, but in the text it is stated to be found in 61 cases (3.4 section). This is an internal inconsistency that must be clarified.
  2. Severity ratings between Lexi-Interact™ and Drugs.com™ are briefly mentioned, but the paper does not provide a detailed comparison table or kappa statistics to quantify agreement between databases. The clinical relevance of “moderate” severity interactions is not explained with concrete patient outcomes. A qualitative example or two would help bridge database classification with actual bedside consequences.
  3. ADRs are reported in 14 patients (17.5%), but it is unclear how many of these were causally linked vs. coincidental with PDDIs. A causality assessment tool (e.g., Naranjo scale) could strengthen validity.
  4. While the paper distinguishes pharmacokinetic vs. pharmacodynamic mechanisms, there is no breakdown of their relative proportions or discussion on which mechanism types were most associated with ADRs.
  5. Minor issues: There is a duplicated phrase: “approved by this study was approved by the Ethics Committee” (line 381–382) — needs grammatical correction. Table 2 includes brand names in parentheses inconsistently; for international readership, brand names may be unnecessary unless relevant to formulation differences.
  Overall I suggest major revision.

Reviewer 2 Report

Comments and Suggestions for Authors

The manuscript reports a prospective, single-center pharmacoepidemiologic study of pDDIS in hospitalized pediatric oncology patients in Iran, screened using two checkers Lexi-Interact™ and Drugs.com™. The topic is relevant, methods are broadly appropriate, and the prospective design and dual-database approach add value. However, several major methodological and reporting issues (below) must be addressed before the paper is suitable for publication. Some comments follow: 

1) Ranitidine has been withdrawn from the market due to concerns about the presence NDMA. Is it not the case in IRAN? And a formulation that it might contain an impurity associated with oncogenesis is administered in cancer patients and peadiatric ones? 

2) Table 4 is excessively detailed and difficult to interpret. The authors could use a venn diagram approach and provide a table of the top ~10 most clinically relevant interactions in the main text while relocating the full exhaustive table to supplementary materials.

3) The phrase “patients underwent dose modification or enhanced monitoring” implies an interventional study. Please clarify if this was described withint the protocol and if the patients (or from the legal guardians) signed for it. Also if it lead in deviations from the chemotherapy protocol and if the treating oncologists were informed etc. 

4) The methods do not specify the tool or algorithm used to attribute adverse drug events (ADEs) as ADRs due to DDIs. If a standardized causality assessment tool (e.g., Naranjo, WHO-UMC) was used, describe it. If not, acknowledge this as a limitation and consider re-assessing events using a validated method.

5) Figure 1. Please clearly define whether Figure 1 presents all side effects (expected pharmacological effects) or ADRs linked to DDIs (via causality assessment).

6) Polypharmacy. The >8 medications threshold should be justified (median value, literature precedent, or institutional definition). Consider adding sensitivity analyses for ≥5 medications, which is a common polypharmacy definition in the literature. This will improve comparability and test robustness.

Reviewer 3 Report

Comments and Suggestions for Authors

A very interesting manuscript entitled ‘Prevalence and Clinical Significance of Potential Drug-Drug Interactions in Hospitalized Pediatric Oncology Patients: A Prospective Pharmacoepidemiologic Study’ was submitted to be considered for its eventual publication in the journal Cancers. Thus, after having read carefully the manuscript, I consider it suitable for its publication practically in its present form due to many positive aspects, but I am not sure if Cancers would be the best choice to do it. Before all, the topic is very important, and, undoubtedly, it would be of high interest for many potential readers. However, Cancers by MDPI is a high impact journal (IF-3034: 4.4) and its broad spectrum of readers expects articles in the edge of knowledge related to cancer. I hope not to cause misunderstanding; the paper has enough merits and positive aspects to be published as I mentioned before (in its present form), but it is limited to a regional scope. Indeed, the data collected for the study comes from a localized hospital. Of course, the main findings are very interesting and important,  but I leave the decision to the Editor. For this reason, I would suggest major revision (not because the article needs it, at all). As the unique suggestion if the paper is accepted, a brief grammar/style revision by an English-native speaker will benefit the reading.